# Debris flows recorded in the Moscardo catchment (Italian Alps) between 1990 and 2019

Lorenzo Marchi[1], Federico Cazorzi[2], Massimo Arattano[3], Sara Cucchiaro[2], Marco Cavalli[1], Stefano Crema[1]

[1] Research Institute for Geo-Hydrological Protection (CNR IRPI), National Research Council of Italy, Corso Stati Uniti 4, 35127 Padova, Italy
[2] Department of Agricultural, Food, Environmental and Animal Sciences, University of Udine, Via delle Scienze 206, 33100 Udine, Italy
[3] Research Institute for Geo-Hydrological Protection (CNR IRPI), National Research Council of Italy, Strada delle Cacce, 73, 10135 Torino, Italy

*Correspondence to*: Lorenzo Marchi (lorenzo.marchi@cnr.it)

**Abstract.** This paper presents debris-flows data recorded in the Moscardo Torrent (eastern Italian Alps) between 1990 and 2019. In this time interval, 30 debris flows were observed, 26 of them were monitored by sensors installed on the channel, while four were only documented through post-event observations. Monitored data consist of debris-flow hydrographs, measured utilizing ultrasonic sensors, and rainfall. Debris flows in the Moscardo Torrent occur from early June to the end of September, with higher frequency in the first part of summer. The paper presents data on triggering rainfall, flow velocity, peak discharge, and volume for the monitored hydrographs. Simplified triangular hydrographs and dimensionless hydrographs were derived to show the basic features of the debris flows in the Moscardo Torrent (time to peak, surge duration, flow depth) and permitting comparison with other instrumented catchments. The dataset is made available to the public with the following
DOI: https://doi.org/10.1594/PANGAEA.919707.

## 1 Introduction

Debris-flow research requires experimental data that are difficult to collect because of the intrinsic characteristics of these processes. Debris flows are locally rare events: although every year they affect several catchments in a given region, their frequency in most channels is usually low (i.e. less than one event per year), and this makes the deployment of instrumentation
not convenient in most potentially affected catchments. The short duration of debris flows, moreover, hampers the possibility of direct observations in channels not equipped with permanent and automatic monitoring devices.

Both post-event field observations and monitoring in instrumented channels are suitable to collect debris-flow data, even if with different resolutions and purposes. Post-event observations enable collecting data (e.g. date of occurrence, deposited volume, traveled distance) at multiple sites or even at the regional scale (Macconi et al., 2008), but permit only the indirect -
and roughly approximate - assessment of important flow variables, such as the flow depth and velocity. Monitoring in instrumented channels enables real-time recording of debris-flow data that cannot be gathered through post-event surveys in

ungauged channels. Given the above-mentioned constraints resulting from the episodic debris-flow occurrence, a careful choice of the sites for monitoring is mandatory, the high frequency of the debris flows being a fundamental requisite to justify the investment.

Starting from early studies in Japan and China (Okuda et al. 1980; Zhang 1993), many papers presented and analyzed debris-flow data collected in instrumented channels; a recent review (Hürlimann et al., 2019) discusses achievements and open problems in debris-flow monitoring.

In many geographical regions, such as, for instance, the European Alps, even in the most active catchments the debris-flow frequency does not exceed one or two events per year. As a consequence, several years are necessary to collect in instrumented

channels debris-flow datasets that are representative of catchment response to different meteorological forcings and variations in sediment availability. The continuation of debris-flow monitoring over multidecadal intervals, in turn, implies changes in the technology of monitoring sensors and data recording and archiving that makes it difficult to collect homogeneous datasets. Implementing consistent datasets through the revision of past data collected in instrumented catchments and making them freely available may contribute to the advance of debris-flow research.

As far as we know, the Moscardo Torrent (Marchi et al., 2002) basin was the first catchment equipped with permanent instrumentation for debris-flow monitoring in Europe. The monitoring activities in the Moscardo Torrent began in 1989-1990 and still keep on, although with some gaps due to the construction of control works in the instrumented channel (1998-2001) and the obsolescence of the instrumentation between 2007 and 2010. Debris-flow monitoring in the Moscardo Torrent was started by the National Research Council of Italy – Research Institute for Geo-hydrological Protection and is continuing, since

2010, in collaboration with the University of Udine.

This work aims to present a dataset of debris flows recorded in the Moscardo Torrent between 1990 and 2019, which were so far unpublished or available from various sources. The following data are presented: date of debris-flow occurrence, triggering rainfall, debris-flow hydrographs, peak discharge, and volume. These data were selected because they define the fundamental characteristics of debris flows, and were recorded employing instrumentation (rain gauges and ultrasonic sensors

for flow stage measurement) installed in the study catchment across the whole duration of the monitoring period. Data recorded by other instruments, namely seismic sensors (Arattano, 1999) and video cameras (Arattano and Marchi, 2000), which are available only for a part of the monitoring period, are not considered here.

**2 The Moscardo Torrent basin**

The Moscardo Torrent basin (Fig. 1, Table 1) is located in the Carnic Alps, in the eastern sector of the Italian Alps. The climate

is temperate with cold winters and abundant precipitation in all seasons; precipitation from November–December to March–April prevailingly occurs as snowfall. Basin slopes are mostly covered by coniferous forest (Norway spruce and silver fir); bare scree and outcropping rocks are present in the upper part of the catchment and along the main channel.

A deep-seated gravitational deformation involves most of the basin, and a large roto-translational landslide occupies the right slope in the middle sector of the catchment (Marcato et al., 2012). The weak rock mass properties and the presence of large amounts of loose debris result in rockfall and shallow landslides that supply large amounts of debris to the channels.

In order to stabilize the channel and reduce the downstream transfer of sediment, 32 check dams and bed sills have been built along the Moscardo channel in the last 40 years: a recent paper by Cucchiaro et al. (2019b) describes the characteristics and temporal evolution of these works and analyses their performance in controlling the debris flows.

The Moscardo Torrent basin was chosen for debris-flow monitoring mainly for the high frequency of such events, while other favorable factors are the easy accessibility of the lower and middle parts of the basin, and the presence of a stable channel on the alluvial fan.

<Figure 1, Table 1>

## 3 Monitoring system and data

Early instrumentation was installed in 1989-1990, and the monitoring activities are still on course, although the time series has some gaps. Table 2 reports basic data on the debris flows observed in the Moscardo Torrent basin. Four debris flows, which were not recorded because of malfunctioning of the instrumentation (1995, 2009, and 2010) or because the sensors had been removed when artificial dikes were built in the monitored channel (1998), were documented through field surveys.

The flow stage at the Moscardo Torrent is recorded using ultrasonic sensors. The sensors are installed in the middle sector of the alluvial fan, where the channel has an average slope of 10%, suspended on a tension structure over the thalweg. The location of the instrumented cross-sections varied during the monitoring period (Fig 1 and Table 3). The recording intervals also varied (Table 3); we underline the relatively coarse recording interval (10 s) of the debris flows observed from 1990 to 1994. Until 1997 the channel on the alluvial fan was in natural conditions, except for a bed sill aimed at protecting a pipeline. Hydraulic works, intended at preventing the overflowing of the alluvial fan were implemented in 1998-2000: the channel was artificially widened and lined with riprap and bed sills, and dikes were built. Figure 2 compares three cross-sections of the channel under natural conditions, surveyed in 1991-1993 with two lined cross-sections surveyed in 2012-2016; these cross-sections are located in the mid-fan area, near the monitoring sites A and D (Fig. 1 and Table 3). Artificial cross-sections are wider than the natural ones; the deposition of lateral levees within the artificial channel, however, led to the restoration of partially natural-like morphological conditions.

One video camera was installed at the monitored cross-section D (Fig. 1); it was initially (1996-1997) installed on the right bank of the channel (Arattano and Grattoni, 2000), and then moved in 2002 to the left bank of the same cross-section. A new video camera was installed in 2016 at the same site. Until 2016 video recordings are available only for a few debris flows; in the most recent years, better night vision and improved reliability of the recording system permitted a more continuous collection of debris-flow videos.

Several debris flows consisted of more than one surge; in these cases, the velocity and peak discharge data reported in Table 2 are related to the main surge. The mean debris-flow velocity was calculated as the ratio of the distance between two instrumented cross-sections to the time difference between the occurrence of the peak of the debris flow in the two recorded hydrographs. The debris-flow volume was computed by summing up, over the entire duration of the event, the product of mean flow velocity and cross-section area occupied by the flow at each time increment. The assumptions underlying this approach to volume computation, and the possible associated errors are discussed in Marchi et al. (2002) and Arattano et al. (2015).

Debris-flow volumes observed between 1990 and 2019 range from 730 m$^3$ to 89500 m$^3$ (Table 2). Marchi et al. (2019) have explored the relationship between catchment area and debris-flow volume in northeastern Italy using quantiles regression. Notwithstanding the large availability of loose debris in the source areas and the abundant precipitation in the Moscardo area, even the largest debris flows observed between 1990 and 2018 lie well below the debris-flow volumes corresponding to the highest percentiles: for the 98[th] percentile, the central value is 195894 m$^3$, with uncertainty bounds between 170902 and 223211 m$^3$. In the Moscardo catchment, the frequent occurrence of debris flows in the monitoring period has probably limited the magnitude of individual events.

Based on video recorded at the monitoring station, the analysis of the hydrographs, and on the observation of the deposits in the monitored channel reach, a few events were classified as debris floods. Post-event observations revealed that also these events occurred as debris flows upstream of the alluvial fan: sediment deposition, also favored by check dams for the event of 24.08.2006 (Arattano et al., 2012), led to transformation into debris floods. No debris floods were observed in the first years of monitoring: the absence of video observations until 1996 could have caused some debris floods to remain unperceived.

<Table 2, Table 3, Figure 2>

The rain gauges installed in the Moscardo basin (Fig. 1) are intended to record the rainfall during the debris-flow season (from late spring to autumn) and are not equipped with heating elements. Table 4 reports their years of operation, elevation, and logging intervals.

<Table 4>

## 4 Summary of recorded data

### 4.1 Rainfall thresholds

After the first nine years of observations (1990-1998), Deganutti et al. (2000) identified a rainfall intensity threshold for debris flow occurrence in the Moscardo catchment. A time interval of at least  6 hours with null or negligible precipitation ($\leq$ 0.2 mm) was chosen for separating the rainfall events (Deganutti et al., 2000). Duration and mean intensity of triggering rainstorms were computed from the onset of precipitation to the passage of the debris flows at the stage measurement stations. This choice implies an approximation in rainfall duration because debris flows are recorded on the alluvial fan (Fig. 1) some minutes after

their initiation in the upper sector of the catchment, but this discrepancy is small if compared to the uncertainties that commonly affect the time of occurrence of landslides and debris flows in the Alps (Palladino et al., 2018).

The threshold defined by Deganutti et al. (2000) is confirmed by the more recent data (Fig. 3a). Moreover, it is possible to identify an upper limit, with the same exponent, for the rainstorms that triggered debris flows in the Moscardo Torrent. Both the lower critical threshold and the upper limit were fitted empirically, a procedure that we consider acceptable due to the relatively small sample size. The two rainfall thresholds have the form:

$$I = a \cdot D^{-0.7} \tag{1}$$

where *a* is 15 for the lower threshold and 50 for the upper limit.

The different rain gauges location during the monitoring period (Fig. 1 and Table 4), and the coarse time resolution in 1990-98 (60' data), which can lead to underestimating the triggering rainfall (Marra, 2019), had a limited impact on the critical rainfall thresholds, which lie in a rather narrow intensity belt.

< Figure 3>

The above-mentioned study by Deganutti et al. (2000) had also shown that debris flows in the Moscardo basin were triggered by rainstorms which had a minimum of 21 mm of total rainfall and at least 60' rainfall intensity of 12.6 mm h$^{-1}$. These thresholds were applied to the months between June and September of the entire precipitation dataset for detecting the rainstorms that have the potential of causing debris flows in the Moscardo (potentially triggering rainstorms), regardless if they caused debris flows. The rain gauges considered were Pramosio (1990-1998), Rio Lares (1999-2006 and 2011-2016), and La Musa (2016-2019). After removing the rainstorms that have caused debris flows, a sample of non-triggering events was obtained and is plotted in the duration-intensity graph of Fig. 3b.

The filter on total rainfall amount and 60' intensity led to exclude several low-intensity precipitation events: 85 non-triggering rainstorms were extracted from the database. It is worth noticing that the automatic extraction of rainstorm events leads to the identification of duration and rainfall quantities that can hardly be compared to the expert-driven event identification. While the expert-based event definition can leverage the availability of debris-flow timing information and unravel the role and importance of rain and hiatuses, the automatic procedure relies only on thresholds of rainfall amount and intensity, showers separation. As such, on average the automatically-extracted rainfall events tend to be longer than the expert-identified ones as they include rainstorm tails.

Most of the non-triggering rainstorms so identified plot above the empirical threshold for debris-flow occurrence, and a few ones (all of them for a duration longer than 10 hours) also above the upper threshold of triggering rainstorms. Non-triggering rainstorms lying above the critical rainfall threshold confirms that, even in a catchment with abundant sediment supply, like the Moscardo, intense rainfall is not the only factor required for debris-flow occurrence. A time interval – even though short – for sediment recharge is probably needed between a debris flow and the next one. The time series of Moscardo, however, shows that some debris flows took place a few days after the previous one (11, 19, and 20 July 1993; 24 and 27 September 2012; 11, 13, and 22 July 2016). Although triggered by different rainstorms and resulting in different debris-flows,

from the point of view of sediment transfer from the source areas to the alluvial fan, these events can be ascribed to the same debris-evacuation episode. A more comprehensive analysis of rainfall-related variables and their interaction with sediment recharge periods between consecutive debris flows (Pastorello et al., 2018), which is outside the presentation of experimental

data proposed in this paper, could shed more light on the processes that control debris-flow occurrence in the Moscardo catchment.

## 4.2 Debris-flow occurrence

The debris flows in the Moscardo Torrent occurred in a time interval of 119 days, from the beginning of summer (4 June 2019) to early autumn (30 September 1991), with 18 out of 30 events occurring in the first 50 days (Fig. 4).

Precipitation data were analyzed to explore a possible control of intense rainfall on the seasonal distribution of the debris flows. In the first years of observation, the rain gauge of Pramosio shows that most of the potentially triggering rainstorms, as defined in the previous section, occurred in June and July (24 rainstorms, versus 15 in August and September). The most recent period, however, shows a much more balanced distribution of intense rainstorms, both at Rio Lares (24 events in June and July versus 27 in August and September), and at La Musa (10 events in June and July versus 9 in August and September). If we consider

the entire observation period (1990-2019), the distribution of potentially triggering rainstorms does not show a disproportionate number of events in the first 50 days, and the occurrence of such storms across the debris-flow season is almost constant (Fig. 4). The absence of a rain-gauge that covers the entire monitoring period prevents more definite conclusions, but the analysis of available rainfall data indicates that the differences in the occurrence of intense rainstorms between June-July and August-September are not closely related to the higher frequency of the debris flows in the first part of summer.

A factor that could explain the more frequent occurrence of debris flows in the first part of summer is the larger availability of debris stored in the channel bed in the debris-flow initiation areas after winter and the snowmelt period. Visual field observations support this interpretation, but the absence of systematic measurements of variations in sediment availability does not permit us to confirm it.

<Figure 4>

Figure 5a shows the days of debris-flow occurrence in the 30 years of observations: the tendency toward the earlier occurrence of debris flows is weak (linear regression coefficient r = - 0.17) and not significant (p-value=0.363).

<Figure 5>

Most debris flows (69%) occurred in the second half of the day, with 9 events from 12 to 18 hours and 9 from 18 to 24 hours: this is consistent with the triggering commonly caused by summer convective rainstorms, which mostly take place in

the afternoon or the evening (Fig. 5b). This result is especially visible in the warmest months, from June to August, whereas 4 out of the 6 debris flows observed in September occurred from midnight to 6 hours. For 4 out of the 30 events, the hour of occurrence is not known: these debris flows were not recorded by the monitoring system and are only known through post-event observations (Table 2).

Figure 6 plots the number of debris flows in each year and the cumulative number of debris flows in the observation period. The absence of debris flows in 2007 and 2008 could be referred to the construction of check dams along the main channel in the middle and lower part of the basin (Cucchiaro et al., 2019b) that trapped most of the material and prevented the debris flows to reach the alluvial fan, but this does not apply to the years 2013-2015. We could argue that the large evacuation of sediment from the catchment in two large debris flows of September 2012 (Table 2) required a few years of sediment recharge before new debris flows were produced. However, the lack of data on the variations of sediment storage in the source areas does not permit us to verify this interpretation. Similar mechanisms could be advocated to explain the absence of debris flows between 1999 and 2001. Artificial channel widening and the construction of bed sills (Fig. 2) could have favored the deposition of some minor flow events, but these factors did not prevent the occurrence of debris flows in the following years. The number of potentially triggering rainstorms in the years 1999-2001, 2005, and 2013-2015, when no debris flows were recorded, was compared with that of the years in which debris flows did occur. The years 2007-2010 were not included in the analysis because no rainfall data are available (Table 4). The mean number of potentially triggering rainstorms in the years without debris flows (2.61), weighted by the number of years of observations at each rain gauge, is lower than in years with debris flows (4.86). However, large variability exists in the number of potentially triggering rainstorms, both in the years with (standard deviation 2.29) and without (standard deviation 2.33) debris flows. The absence of debris flows in some years in an active catchment like the Moscardo is probably caused by a combination of extrinsic, weather-related factors, i.e. the occurrence of a small number of intense rainstorms, and factors intrinsic to the catchment, i.e. the effect of control works and temporary scarcity of mobilizable debris.

We underline that our record consists of debris flows that reached the alluvial fan and were recorded there by the installed instrumentation, or documented through post-event surveys. Field observations, although not systematic, have provided evidence of other flow events that did not reach the alluvial fan (for instance, on July 20 and August 20, 2001), not even as debris floods, but the record of these small events is not complete.

<Figure 6>

The mean frequency between 1990 and 2019 is 1.0 event per year including three debris floods (Table 2). The frequency is similar or lower than other instrumented basins in the Alps if the monitoring stations at the basin outlet or on the alluvial fan are considered: Réal (France): 1.0 event/year (Hürlimann et al., 2019), Illgraben (Switzerland): 3-5 debris flow every year plus some debris floods (Hürlimann et al., 2019), Gadria (Italy): 1.3 events/year between 2011 and 2017 (Comiti et al., 2014; Coviello et al., 2020).

## 4.3 Debris-flow hydrographs

The analysis of the debris-flow hydrographs permitted to define their basic characteristics, such as maximum flow depth, time to peak, and surges duration. These characteristics have been used to derive triangular hydrographs and dimensionless hydrographs that could be employed to define realistic inputs for debris-flow mathematical models. Such hydrographs could also allow comparisons with hydrographs recorded in other instrumented catchments.

The analysis was performed examining individual debris-flow surges. In the Moscardo, the water level before the occurrence of a debris flow is negligible if compared to the maximum depth of debris-flow surges: the start of a surge is easily identified at the first rise of the hydrograph. Some uncertainties may arise regarding the end of the recession phase, which occurs when the flow level becomes almost stable or a new surge begins. Data from 62 surges were collected, i.e. on average approximately two surges per debris-flow event. The maximum flow depth was computed as the difference between the surge's peak and the level before the start of the flow rise. A minimum rise of 0.5 m was adopted for the analysis: this value is a trade-off between the need of considering all relevant flow surges and the requirement of minimizing the risk of including in the analysis minor level fluctuations that are not representative of the debris-flow behavior in the studied channel. Table 5 reports basic statistics on the time to peak, the duration of the recession, and the maximum flow depth. In six cases, while the rising limb and the debris-flow peak are clearly defined and allowed the calculation of time to peak and maximum flow depth, the duration of the recession phase could not be properly recognized (for instance, because of substantial channel aggradation).

<Table 5>

We have devised, based on the data of Table 5, the triangular hydrographs presented in Figure 7 to provide a simplified representation of the debris-flow hydrographs in the downstream reach of the Moscardo Torrent. We used the summary data related to duration and the maximum depth of the debris-flow surges to define three triangular hydrographs related to different levels of event severity:

a) a hydrograph characterized by the median value of the time to peak, flow depth, and recession duration;

b) a severe event, coupling a short time to peak, high flow depth, and long duration (25th percentile for the time to peak and 75th percentile for flow depth and recession duration);

c) a low-severity event, featuring a relatively long time to peak low flow depth and short recession (75th percentile for the time to peak and 25th percentile for flow depth and recession duration).

The hydrograph "b" features fast arrival of the front, high flow depth, and long duration (hence high peak discharge and volume too): all these characteristics define a challenging debris-flow event. Opposite conditions characterize the low-severity hydrograph "c". The synthetic representation provided by Figure 8 enables us to define, based on a simple statistical analysis of the data collected, the worst scenario expected in the studied channel, as far as the maximum flow depth and the shortest time to peak of debris flows are concerned. In the absence of mathematical modeling that could provide more in-depth information, this can be of some use for the planning of mitigation measures.

<Figure 7>

A more detailed representation of hydrographs shape was achieved by averaging the recorded hydrographs of the debris-flow surges. This analysis was performed on the debris flows recorded between 2002 and 2019: data for 12 surges for both the upstream (E, Fig. 1) and downstream (D) measuring stations were available. The debris flows recorded between 1990 and 1998 were excluded because of coarser time resolution from 1990 to 1994 and variable cross-sectional geometry. Dimensionless hydrographs were generated normalizing the flow depth by its maximum value and the time by the total surge duration. The flow peaks were aligned to preserve the sharp shape that is a distinctive feature of debris-flow hydrographs.

Finally, the ordinates were averaged, and mean debris-flow hydrographs were obtained, one for each measuring station (Fig. 8). Since the duration varies from surge to surge, average debris-flow hydrographs were computed only for the interval covered by at least 50% of the hydrographs, which led to consider six hydrographs for both the considered measuring cross-sections. The choice of averaging the hydrographs on the interval covered by at least 50% of the events is intended to limit the impact on the results of the peculiar characteristics of single surges.

<Figure 8>

The dimensionless hydrographs at the two measuring stations, which are quite close to each other (Fig. 1) and have similar cross-sectional geometry display similar shapes (Fig. 9). It can be noted that the precursory surge that precedes the peak is somewhat larger in the upstream monitoring station, whereas the ordinates of the intermediate part of the recession phase are higher in the downstream cross-sections and lower in the last part of the recession limb of the hydrograph. These differences could indicate that flow propagation, even in a short (76 m) channel reach, causes non-negligible changes in hydrographs' shape. In our particular case, the hydrograph deformation, due to the downstream propagation of the debris flow wave along the channel reach between the stations, does not lead to a spreading of the hydrograph but its shrinkage. It would be interesting to verify if this behavior, which is in contrast to the hypothesis that debris flows behave as kinematic waves, will be confirmed by further observations in this torrent and will be also observed in other catchments.

## 5 Conclusions

A 30 years dataset of debris-flows recorded in an instrumented basin of the eastern Italian Alps has been presented. During the monitoring period, data collection and analysis has been conditioned by variability in financial resources, and by the implementation of hydraulic works, which caused temporary disruption of data recording. Further disturbance to the operation of the monitoring system came from the malfunctioning and obsolescence of the sensors (especially the ultrasonic sensors), power outages, and damage caused by wind storms. Notwithstanding these problems, which often affect field monitoring activities, particularly if they are carried on for a long period, the monitoring installations allowed recording 26 out of the 30 debris flows that reached the alluvial fan between 1990 and 2019. For the four remaining events, at least the date of occurrence is known.

The relatively large number of debris-flow hydrographs recorded by the ultrasonic sensors has permitted deriving simplified triangular hydrographs that show the distinctive features of debris flows (short total event duration and a very short time to peak). Based on the values of peak flow depth, time to peak, and total surge duration, three triangular hydrographs were devised that correspond to different event severity. This representation, which could be adopted elsewhere for comparison, provides a quick view of possible debris-flow responses and may help to define realistic inputs to debris-flow propagation models.

A more detailed representation of the shape of the hydrographs was also achieved by averaging the recorded hydrographs of the most representative debris-flow surges and generating dimensionless hydrographs, one for each gauging station, through

the normalization of the flow depth by its maximum value and the time by the total surge duration. These dimensionless hydrographs provide a characterization of the shape of the hydrographs at the two measuring stations in the studied channel.
This may enable comparison with the monitoring results obtained in other instrumented catchments, allowing to identify similarities and differences and relate them to basin and channel characteristics. Finally, it could also permit the study of the hydrograph deformation along the torrent and provide insights on the dynamical behavior of debris flows.

The Moscardo Torrent dataset could contribute to further analysis, in addition to those already carried out and reported in previous studies and those briefly outlined in this paper. We mention here the comparison of triggering rainfall and basic flow
variables (depth, velocity, volumes) with other basins instrumented for debris-flows monitoring under different climate and geolithological conditions (Hürlimann et al., 2019).

Some limitations that arise from data presentation need to be mentioned. Rainfall data have been recorded at various rain gauges located at different elevations within or near the catchment, but a time series of precipitation recorded at the same rain gauge for the whole monitoring period is not available. Changes in the channel topography caused by hydraulic works, as well
as and the rather low recording interval of the hydrographs until 1994, limit the use of numerical models for the reanalysis of the debris flows recorded in the first years of monitoring. Another problem is that debris-flow monitoring was not fully matched by systematic topographic surveys that permit assessing the variation of sediment stored along the channels and the amounts of erodible debris in the source areas connected to the channel network. Beginning in 2011 (Blasone et al., 2014), Terrestrial Laser Scanner (TLS) and Structure from Motion (SfM) topographic surveys carried out in selected sectors of the basin, have
permitted evaluating the erosion and depositions related to debris flows (Cucchiaro et al., 2018)  and the effect of check dams on sediment dynamics (Cucchiaro et al., 2019a). The lack of this information for the whole basin and the entire monitoring period, however, has so far hampered the understanding of the variations in frequency and magnitude of the debris flows in the Moscardo Torrent basin. This issue could be solved if more systematic topographic surveys of the sediment source areas will be carried out in the future: the progress in SfM techniques and UAV surveying could enable a more frequent collection
of topographic data also in unstable areas of difficult access like the headwater of the Moscardo Torrent.

*Data availability.* Debris-flow data presented in this paper are available at the following DOI: https://doi.org/10.1594/PANGAEA.919707.

*Authors contribution.* LM initiated and elaborated this work, and wrote most of the paper. All authors contributed to the paper's preparation and participated in the implementation of the monitoring system and the collection of field data.

*Competing interests.* The authors declare that they have no conflict of interest.

*Acknowledgements.* The Regione Autonoma Friuli Venezia Giulia collaborated on the installation of the monitoring system and data collection. The authors wish to thank Marcel Hürlimnann and one anonymous reviewer for their constructive comments on the first version of this paper.

*Financial support.* The monitoring activities in the Moscardo catchment are carried on in the frame of an EU Interreg V-A
Italy-Austria 2014-2020 project (ITAT3035 – INADEF). This paper was supported by the project DTA.AD003.474 "Cambiamento climatico: mitigazione del rischio per uno sviluppo sostenibile", funded by the Italian National Research Council.

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

## Figures and tables

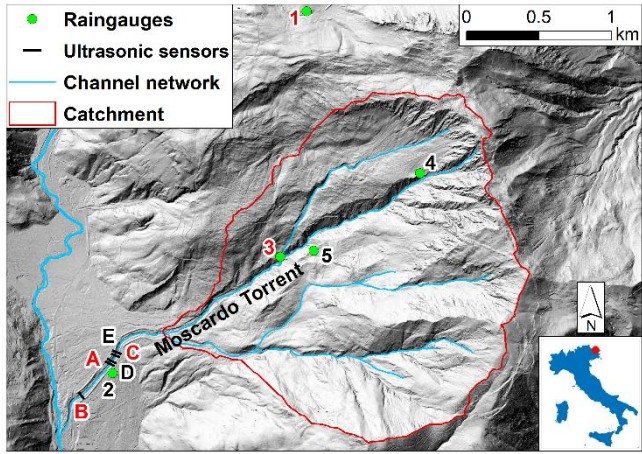

**Figure 1. The Moscardo Torrent basin (DTM of Friuli Venezia Giulia Region). The letters and numbers in red refer to monitoring sites that are no more active.**

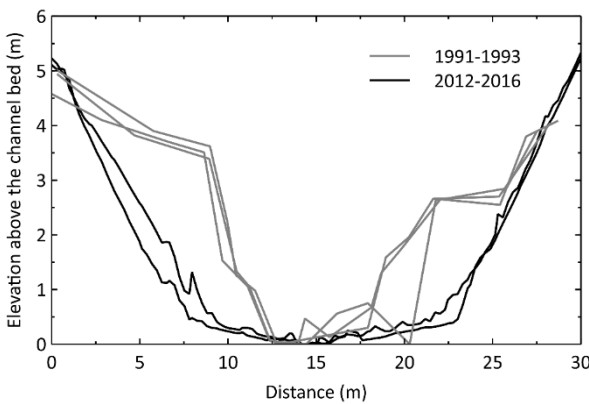

**Figure 2. Cross-sections in the mid-fan area (near the monitoring sites A and D, Fig. 1) before and after the implementation of control works in 1998-2000.**

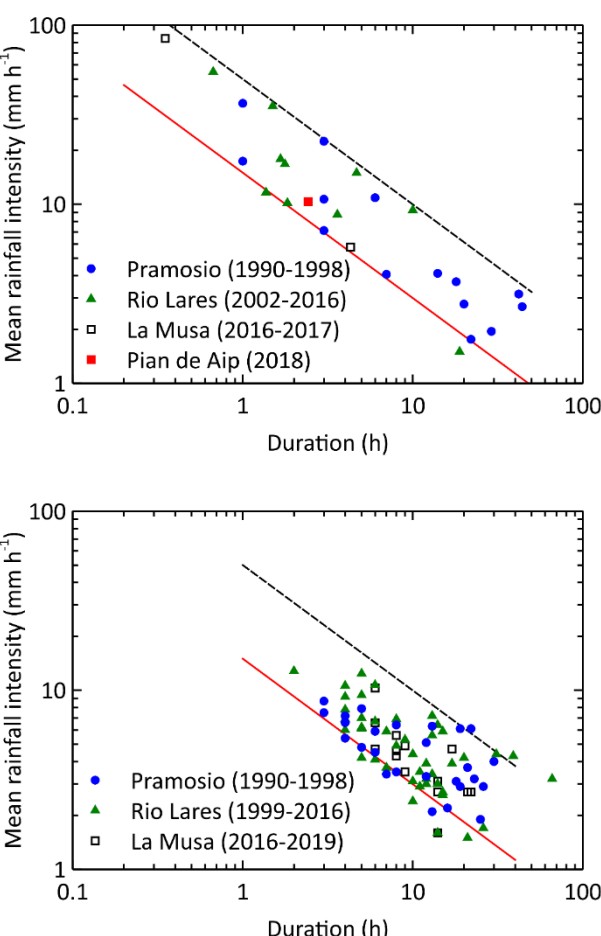

**Figure 3. (a) Plot of mean rainfall intensity versus duration for debris-flow triggering rainstorms; the rain gauge of Pian de Aip was used for the 2018 debris flow because of a gap in data in the La Musa rain gauge. (b) Plot of mean rainfall intensity versus duration for high-intensity rainstorms that did not trigger debris-flows.**


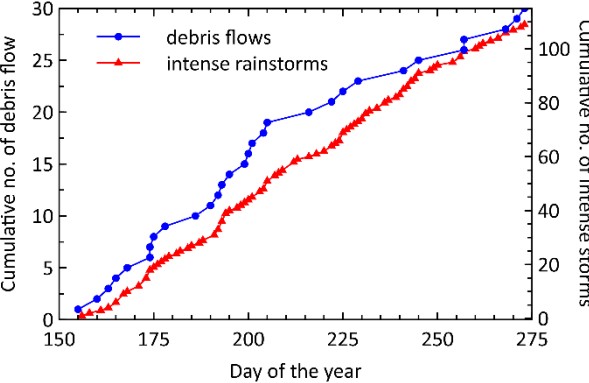

**Figure 4. Day of occurrence versus cumulative number of debris flows and intense (potentially triggering) rainstorms in the observation period.**

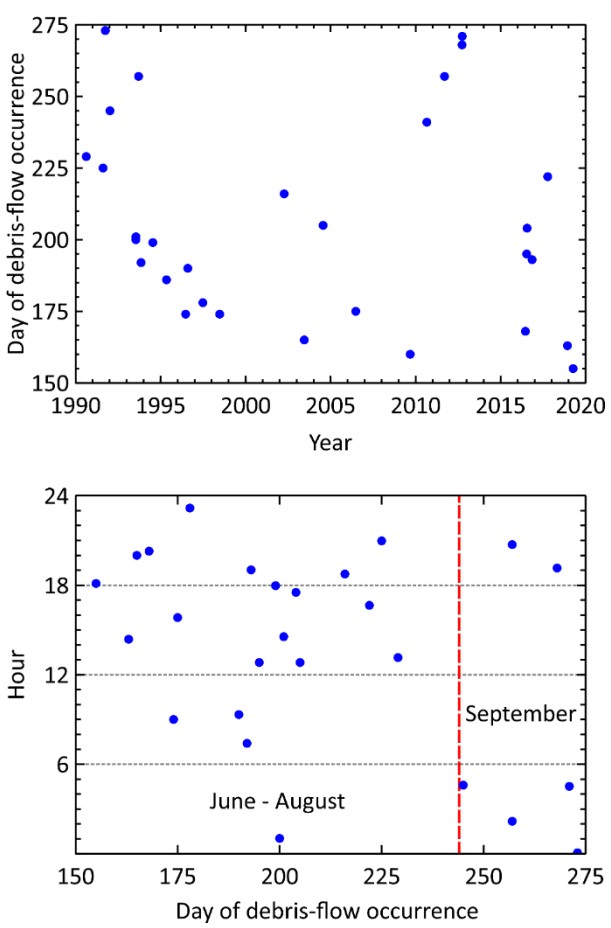


**Figure 5. (a) Days of debris-flow occurrence between 1990 and 2019. (b) Hours of debris-flow occurrence.**

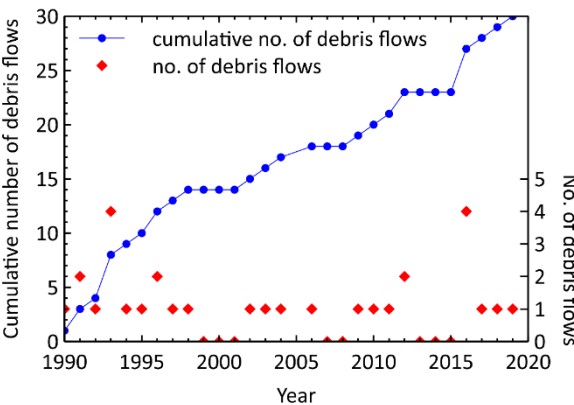

**Figure 6. Number of debris flows occurred in each year and cumulative number of events in the observation period. There are no years with missing data: when the monitoring instrumentation was not working, debris-flow occurrence was documented through field observations.**

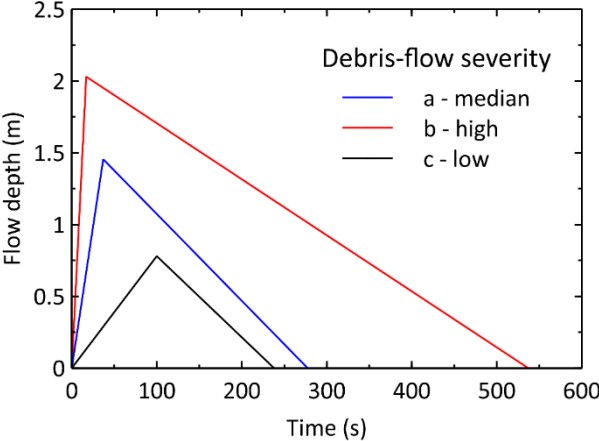

**Figure 7. Simplified triangular debris-flow hydrographs derived from debris-flow surges recorded from 1990 to 2019.**

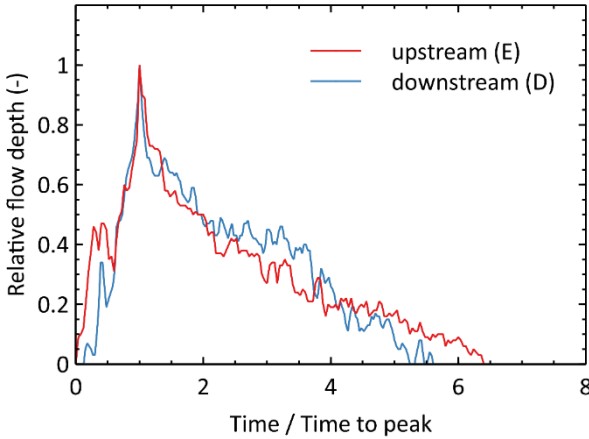

**Figure 8. Dimensionless debris-flow hydrographs (2002-2019) The letters (E) and (D) refer to the instrumented cross-sections shown in Fig. 1.**


**Table 1. Basic characteristics of the Moscardo basin.**

| | |
|---|---|
| Basin area (km$^2$) | 4.1 |
| Range in elevation (max elevation – fan apex) (m) | 2043 - 890 |
| Mean basin slope (%, degrees) | 63, 32.2 |
| Mean channel slope (%, degrees) | 37, 20.3 |
| Geology | Carboniferous Flysch (Venturini, 2002) |
| Mean annual precipitation (mm) | 1820 rain gauge 1 (Fig. 1), years 2010-2019 (data Regione Friuli Venezia Giulia) |

**Table 2. Debris-flow data; the mean velocity refers to the main surge.**

| No. | Event date | No. of surges (upstream, downstream) | Mean velocity (m s$^{-1}$) | Peak discharge (m$^3$ s$^{-1}$) | Volume (m$^3$) | Previous studies |
|-----|-----------|-------------------------------------|---------------------------|----------------------------------|----------------|------------------|
| 1 | 17.08.1990 | 1, - | 1.0 | - | - | Marchi et al. (2002) |
| 2 | 13.08.1991 | 1, 1 | 5.0 | 88 | 19000 | Marchi et al. (2002) |
| 3 | 30.09.1991 | 1, 1 | 1.9 | 24 | 3250 | Marchi et al. (2002) |
| 4 | 01.09.1992 | 2, 2 | 2.5 | 46 | 5800 | Marchi et al. (2002) |
| 5 | 11.07.1993 | 1, 1 | 3.0 | 14 | 5600 | Marchi et al. (2002) |
| 6 | 19.07.1993 | 1, 1 | 0.9 | 3 | 730 | Marchi et al. (2002) |
| 7 | 20.07.1993 | 1, 1 | 4.3 | 16 | 6500 | Marchi et al. (2002) |
| 8[)] | 14.09.1993 | 1, 1 | 2.5 | - | 3800 | Marchi et al. (2002) |
| 9 | 18.07.1994 | 2, 1 | 4.0 | - | - | Marchi et al. (2002) |
| 10 [a] | 05.07.1995 | - | - | - | - | Marchi et al. (2002) |
| 11 | 22.06.1996 | 3, 3 [b] | 3.5 | 139 | 16133 | Marchi et al. (2002) |
| 12 | 08.07.1996 | 1, 1 | 4 | 194 | 57800 | Marchi et al. (2002) |
| 13 | 27.06.1997 | 1, 1 [b] | 2.9 | 25 | 3000 | Marchi et al. (2002) |
| 14 [a] | 23.06.1998 | - | - | - | 51000 | - |
| 15 | 04.08.2002 | 2, 2 | - | - | | - |
| 16 [c] | 14.06.2003 | - | | | | - |
| 17 [d] | 23.07.2004 | 1, 1 | 5.4 | - | | Arattano and Marchi (2005) |
| 18 [c] | 24.08.2006 | - | 1.6 | - | 5500 | Arattano et al. (2012) |
| 19 [a] | 09.06.2009 | - | - | - | | - |
| 20 [a] | 29.08.2010 | - | - | - | | - |
| 21 | 14.09.2011 [e] | 2, 2 | 3.6 | 71 | 4700[)] | Blasone et al. (2014) |
| 22 | 24.09.2012 | -, 1 | 3-4 [f] | 91-121 | 57000 | Blasone et al. (2014) |
| 23 | 27.09.2012 | -, 3 | 3-4 [f] | 119-159 | 89500 | Blasone et al. (2014) |
| 24 | 16.06.2016 | 1, 1 | 4.5 | 53-87 | 15936 | Cucchiaro et al. (2019a) |
| 25 | 11.07.2016 | 1, 1 | 0.5 | 2-3 | - | Cucchiaro et al. (2019a) |
| 26 [c] | 13.07.2016 | - | 2.4 | 22 | - | Cucchiaro et al. (2019a) |
| 27 | 22.07.2016 | 1, 1 | 4.8 | 95-130 | 21808 | Cucchiaro et al. (2019a) |
| 28 | 10.08.2017 | 2, 1 | 4.0 | 61-94 | 30000 | Cucchiaro et al. (2019a) |
| 29 | 12.06.2018 | 2, 1 | 2.0 | - | - | - |
| 30 | 04.06.2019 | 1, 1 | 0.31 | - | - | - |

[a] No monitoring data, only post-event surveys

[b] Intermediate ultrasonic sensor: 3 surges

[c] Intermediate ultrasonic sensor: 1 surge

[c] Debris flood

[d] See Arattano and Marchi (2005) for the velocity computed using cross-correlation

[e] Incomplete hydrograph

[f] Flow velocity estimated from previous events

**Table 3. Basic characteristics of the debris-flow monitoring system; the letters in the second column refer to the instrumented cross-sections shown in Figure 1.**

| Years | No. of ultrasonic sensors | Length of the instrumented channel reach (m) | Recording interval (s) | No. of recorded debris flows |
|---|---|---|---|---|
| 1990-1994 | 2 (A-B) | 300 | 10 | 9 |
| 1996-1997 | 3 (C-D, D-B) | 370 | 1 | 3 |
| 2001-2006 | 2 (E-D) | 76 | 1 | 4 |
| 2011-2012 | 2 (E-D) | 76 | 2 | 3 |
| 2013-2019 | 2 (E-D) | 76 | 1 | 7 |

**Table 4. Rain gauges in the Moscardo basin: years of operation and recording intervals. The rain gauges numbers in the second column refer to the map of Figure 1.**

| Rain gauge (years of operation) | Code (Fig. 1) | Elevation (m) | 1 hour | 10 minutes | 2 minutes | 1 minute | Event recording |
|---|---|---|---|---|---|---|---|
| Pramosio (1990-1998) | 1 | 1522 | ● | | | | |
| Rio Lares (1998-2000) | 3 | 1081 | | | ● | | |
| Rio Lares (2001-2006; 2011-2012) | 3 | 1081 | | ● | | | |
| Rio Lares (2013-2016) | 3 | 1081 | | | | | ● |
| Shelter (2013-2019) | 2 | 839 | | | | ● | |
| La Musa (2012-2019) | 4 | 1560 | | | | | ● |
| Pian de Aip (2017-2019) | 5 | 1194 | | | | | ● |


**Table 5. Basic statistics on debris-flow surges duration and flow depth.**

| | Valid N | Mean | Std. Dev. | Median | Lower quartile | Upper quartile |
|---|---|---|---|---|---|---|
| Time to peak (s) | 62 | 68 | 76 | 44 | 17 | 100 |
| Recession duration (s) | 56 | 397 | 412 | 234 | 138 | 520 |
| Max flow depth (m) | 62 | 1.48 | 0.76 | 1.46 | 0.78 | 2.03 |