# Peer review of "Debris flows recorded in the Moscardo catchment (Italian Alps) between 1990 and 2019"

_Natural Hazards and Earth System Sciences, 2020_

## Referee Comment (RC1) · Marcel Hürlimann (Referee) · 8 Sep 2020

General comments The manuscript presents data from the debris-flows monitoring system in the Mosardo catchment, which seems to be the oldest in Europe (monitoring over 30 years!). The topic of the ms is perfectly fitting with the themes of the journal and the outcomes are relevant for researchers and practitioners. However, the ms needs some improvements before publications in NHESS. In the following, the major and minor critiques are listed.

Major critiques: I. A general, but important critique is that the explanations and descriptions are in some parts of the ms too short. This lack of complete information makes the understanding of some outcomes a bit complicate. I will describe the parts that

need to be enlarged in the major and minor critiques.

II. The text of some sections is sometimes mixed up and the authors should follow the defined structure or adapt the structure and titles. First example: the contents of Sections 2 (Settings) and 3 (Data): L74-82 should be placed into Section 2, while L65-66 may be stated at the beginning of section 3. Another example is between section 4.1 (occurrence) and 4.2 (rainfall), where the rainfall is already analysed in section 4.1. In addition, I propose changing the title of 4.2 into "Rainfall threshold" (or similar).

III. The relation between rainfall characteristics, sediment availability and debris-flow triggering may be better explored. Detailed data on the sediment availability are not available, but it may be approximated indirectly by number of days between two debris-flow events, volume of previous event etc. Finally, this information should be analysed together with the rainfall characteristics. A similar approach was applied in our monitoring site in the Pyrenees (see Pastorello, R.; Hürlimann, M.; D'Agostino, V. (2018) Correlation between the rainfall, sediment recharge and triggering of torrential flows in the Rebaixader catchment (Pyrenees, Spain). Landslides. 15(10),1921-1934)

IV. The definition of rainfall thresholds is a complex task. The section regarding this topic is very short and more information is necessary of the method how the two thresholds were defined (which rain gauges, how the rainfall duration was determined, how the curves were finally defined etc.). In addition, non-triggering rainstorms must also be added in the plot and commented in the text (explain false positive, false negative etc.). In conclusion, I strongly recommend to improve this part of the ms and enlarge the text.

V. Some Figures need to be improved since information is lacking (legend and more detailed figure captions: see comments below). On the other side, Fig3 and maybe Fig2 are not really substantial and do not refer to the main topics of the ms (debris flow occurrence, rainfall characteristics, hydrographs). I propose including some additional plots on these three topics and maybe delete Figure 3.

Minor critiques: 1. Introduction may be enlarged including some additional information, experiences and open questions of debris-flow research and in particular of instrumental monitoring of debris flows.

2. The title of section 3 may be changed into "Monitoring system and data" (or similar). I propose adding technical details on the ultrasonic sensors and rain gauges used during the last 30 years and some experiences gathered.

3. L83-84: the information on the number of surges would be helpful and should be added in Table 2.

4. L103-118 and Fig.4: add legend in Fig.4. Explain, which rain gauge was used to draw the plot of the potential triggering rainstorms. If the plot includes multiples or all rain gauges, then you have to explain, what was the procedure to avoid duplications. In general, I recommend to better explain the text between L103-118 (especially the last part).

5. L119-122: Good arguments. You may propose some ideas to resolve this aspect. See point III in major critiques.

6. L127: you may create a plot of the time of triggering and add it as Fig. 5b

7. L132-140: This part should be at the beginning of Section 4.1. Afterwards, I would start with the rainfall analysis

8. L185-193 (Fig.8 and Table2): the analysis of the hydrographs is very interesting. I have two suggestions: i) could you provide the return period of three hydrographs? ii) Is it possible to also add the statistics of the surge volumes in Table 2?

Specific comments:

L60. Could you be more precise and replace "several"

L74-75: English is not very clear (to me).

[Figure]

L80 and Fig.2: please add "near the monitoring sites A and D" in the caption of Fig.2

L88 and Fig.1: please add the position of the video camera in Figure 1.

Fig. 7. Add legend

L168: please correct the citation format

L207 and Fig9: please add the cross-section labels D and E in the text and in the plot. This would clarify the actual names (up/downstream) in the plot.

Table 1: please also add the slope angle in degrees

––––––––––––––––––––––––––––

---

## Referee Comment (RC2) · Anonymous Referee #2 · 16 Oct 2020

Informative description of the monitoring situation of the Moscardo torrent.

Comments: It would be fine to get some background information about the instrumentation before chapter 3 (debris flow data). The main recordings are along the cannel are flow height and seismic amplitude. How these data are observed, recorded and treated? Within the sampling interval (measuring interval?) is the record averaged, or is it the maximum etc.? How to start and end a debris flow surge with a Zero value in order to calculate a discharge? How to define the real flow section of a debris flow, if there is only a punctual measurement? How big are possible uncertainties within the dataset? Are there any suggestions?

Line 83... (surge) velocity, mean velocity (see table 2): How is mean velocity calculated, which difference is calculated to (surge) velocity? How is the peak discharge really

estimated?

Ch 4 (Line 100 ) ...beginning of summer (2019) to early autumn (1991). What does this mean? Are the triggering rainstorms independent from the gauging station?

Fig. 4 Legend is missing Are the data shown for all stations? What does 150 and 275 present? Give the information about the day.

Fig. 5: There is no significant regression! Why to present a regression? It is better to show the scattered data.

L135 ...evacuation of sediment.. better: mobilization of sediment

Ch 4.2: For the reader it would be better to combine this Chapter with Ch 4.1 (Ocurrence). Well, Ch 4.1 shows the distributiion of df ocurrence during the year and Ch 4.2 is focused on the precipitation thresholds, but there should be a link between the chapters to come out with some new findings. How is the duration of triggering rainfall defined? Is it the time before the debris flow arrives at the station or less?

Table 1: Just a question: How do we define a catchment (area)? It seems that this area is calculated as the area of the drainage basin (which is hydrological defined). Usually a catchment area includes the are of the fan, too. (see Fig.1) How is mean basin slope and mean channel slope calculated?

Table 2: mean velocity ???? (see above)

Fig. 6: Please include the years of missing data in a different way, not only showing a Zero-value.

Fig. 7: Legend is missing

---

## Author Comment (AC1) · 11 Nov 2020

**Response to Referee 1 - Marcel Hürlimann**

General comments The manuscript presents data from the debris-flows monitoring system in the Mosardo catchment, which seems to be the oldest in Europe (monitoring over 30 years!). The topic of the ms is perfectly fitting with the themes of the journal and the outcomes are relevant for researchers and practitioners. However, the ms needs some improvements before publications in NHESS. In the following, the major and minor critiques are listed.

*We wish to thank Dr. Hürlimann for his constructive comments. Below we present our responses. Our responses includes the changes that we will implement in the manuscript if we will be invited to submit a revised version.*

Major critiques:

I. A general, but important critique is that the explanations and descriptions are in some parts of the ms too short. This lack of complete information makes the understanding of some outcomes a bit complicate. I will describe the parts that need to be enlarged in the major and minor critiques.

*The aim of this paper is to present a catalogue of debris-flow events recorded in an instrumented basin. For this reason, data analysis is focused on a few selected issues and is essentially intended to describe the basic features of the recorded debris flows (date of occurrence, triggering rainfall, and hydrographs shape). Following the suggestions of the reviewer, however, we have extended some parts of the manuscript.*

II. The text of some sections is sometimes mixed up and the authors should follow the defined structure or adapt the structure and titles. First example: the contents of Sections 2 (Settings) and 3 (Data): L74-82 should be placed into Section 2, while L65-66 may be stated at the beginning of section 3. Another example is between section 4.1 (occurrence) and 4.2 (rainfall), where the rainfall is already analysed in section 4.1.In addition, I propose changing the title of 4.2 into "Rainfall threshold" (or similar).

*We accepted the suggestion (no. 2 of Minor critiques) to change the title of Section 3 to include the monitoring system. We kept in this section the text at the lines 74-82, which describes the monitoring system. We moved the text at the lines 65-66 to the beginning of Section 3.*

*We modified the structure of Section 4 (Summary of recorded data). The first subsection presents the rainfall thresholds for debris-flow initiation, while the second and the third subsections deal with debris-flow occurrence (day, hour, etc.) and debris-flow hydrographs, respectively.*

III. The relation between rainfall characteristics, sediment availability and debris-flow triggering may be better explored. Detailed data on the sediment availability are not available, but it may be approximated indirectly by number of days between two debris-flow events, volume of previous event etc. Finally, this information should be analyzed together with the rainfall characteristics. A similar approach was applied in our monitoring site in the Pyrenees (see Pastorello, R.; Hürlimann, M.; D'Agostino, V. (2018). Correlation

between the rainfall, sediment recharge and triggering of torrential flows in the Rebaixader catchment (Pyrenees, Spain). Landslides. 15(10),1921-1934).

*The point raised by the reviewer is undoubtedly relevant. However, according to the aims of this paper, which is intended to present a debris-flow dataset, we would prefer not to explore it. The identification of proxies for sediment availability and their possible influence on rainfall thresholds for debris-flow triggering could become the objective of future studies: we mention this issue in the section on rainfall thresholds.*

IV. The definition of rainfall thresholds is a complex task.  The section regarding this topic is very short and more information is necessary of the method how the two thresholds were defined (which rain gauges, how the rainfall duration was determined, how the curves were finally defined etc.).  In addition, non-triggering rainstorms must also be added in the plot and commented in the text (explain false positive, false negative etc.).  In conclusion, I strongly recommend to improve this part of the ms and enlarge the text.

*We have revised the subsection on rainfall thresholds (now subsection 4.1). We have provided details on the separation of rainstorms, the rain gauges used, and how rainfall duration was determined. We added a plot of rainfall intensity versus duration for non-triggering rainstorms.*

V. Some Figures need to be improved since information is lacking (legend and more detailed figure captions:  see comments below).  On the other side, Fig3 and maybe Fig2 are not really substantial and do not refer to the main topics of the ms (debris flow occurrence, rainfall characteristics, hydrographs). I propose including some additional plots on these three topics and maybe delete Figure 3.

*We removed the figure 3, as suggested by the reviewer. We would prefer to keep the figure 2 because it shows the cross-sectional geometry of the instrumented channel and its variations during the monitoring period. This is an information that relates to the debris-flow hydrographs presented in this paper.*

*We added two figures (non-triggering rainstorms in the rainfall intensity – duration plot), and hour of occurrence versus day of the year, as suggested by the reviewer.*

Minor critiques:
1. Introduction may be enlarged including some additional information, experiences and open questions of debris-flow research and in particular of instrumental monitoring of debris flows.

*We refer to a recent review paper (Hürlimann et al., 2019) for experiences and open questions in debris-flow monitoring. We have extended the introduction by stressing the problems of debris-flow data collection resulting from the low frequency of such events even in the most active catchments and the importance of making the datasets freely available.*

2. The title of section 3 may be changed into "Monitoring system and data" (or similar).I propose adding technical details on the ultrasonic sensors and rain gauges used during the last 30 years and some experiences gathered.

*We modified the title of Section 3 according to the suggestion of the reviewer.*

*Unfortunately we cannot provide technical details on ultrasonic sensors and rain gauges because these instruments were replaced several times and no track was kept of their technical specifications.*

3.  L83-84: the information on the number of surges would be helpful and should be added in Table 2.
*Done. Thank you for this suggestion.*

4.  L103-118 and Fig.4: add legend in Fig.4. Explain, which rain gauge was used to draw the plot of the potential triggering rainstorms. If the plot includes multiples or all rain gauges, then you have to explain, what was the procedure to avoid duplications. In general, I recommend to better explain the text between L103-118 (especially the last part).
*Legend in Fig. 4: done.*
*The plot of rainfall intensity versus duration does not include multiple rain gauges.*

5.  L119-122: Good arguments. You may propose some ideas to resolve this aspect. See point III in major critiques.
*In this paper we comment the existing Moscardo dataset and we mention the absence of measurements of the variations in sediment availability as a limitation of data so far collected. We added a sentence in the conclusions stating that this issue could be solved if more systematic topographic surveys of the sediment source areas will be carried out in the future.*

6. L127: you may create a plot of the time of triggering and add it as Fig. 5b
*Done; thank you for the suggestion.*

7.  L132-140: This part should be at the beginning of Section 4.1. Afterwards, I would start with the rainfall analysis
*We have substantially modified Section 4 (Summary of recorded data): now it starts with rainfall thresholds analysis.*

8.  L185-193 (Fig.8 and Table2): the analysis of the hydrographs is very interesting. I have two suggestions: i) could you provide the return period of three hydrographs? ii)Is it possible to also add the statistics of the surge volumes in Table 2?
*i) The return period likely refers to the peak discharge. We would prefer not to perform this analysis because the small sample size makes such an estimation highly uncertain.*
*ii) We added in the text (section 3) a comment on debris-flow volumes, with a focus on the largest values. Basic statistics of debris-flow volume – as well as of flow velocity and peak discharge – can easily be derived from the data of Table 2.*

Specific comments:

L60. Could you be more precise and replace "several"
*Done: 32 check dams and bed sills.*

L74-75: English is not very clear (to me).
*We have rephrased the sentences at the lines 74-75.*

L80 and Fig.2: please add "near the monitoring sites A and D" in the caption of Fig.2
*Done.*

L88 and Fig.1: please add the position of the video camera in Figure 1.
*We added two sentences on video camera installation and video recordings.*

Fig. 7. Add legend
*Done.*

168: please correct the citation format
*Done.*

L207 and Fig9: please add the cross-section labels D and E in the text and in the plot. This would clarify the actual names (up/downstream) in the plot.
*Done.*

Table 1: please also add the slope angle in degrees
*Done.*

---

## Author Comment (AC2) · 11 Nov 2020

**Response to Referee 2 - Anonymous**

Informative description of the monitoring situation of the Moscardo torrent.
*We wish to thank the reviewer for his/her comments. Below we present our responses.*

Comments: It would be fine to get some background information about the instrumentation before chapter 3 (debris flow data). The main recordings are along the cannel are flow height and seismic amplitude. How these data are observed, recorded and treated? Within the sampling interval (measuring interval?) is the record averaged, or is it the maximum etc.? How to start and end a debris flow surge with a Zero value in order to calculate a discharge? How to define the real flow section of a debris flow, if there is only a punctual measurement? How big are possible uncertainties within the dataset? Are there any suggestions?
*Seismic amplitude is not considered in this paper: we focus on rainfall data and flow stage data because, as we remind in the Introduction, other data are available only for a part of the monitoring period.*
*The systems for flow stage data recording varied during the monitoring period: in the present installation, data are recorded by a Campbell CR1000 data logger. We would prefer not to provide details about present and past recording systems because we think they would be of limited interest to possible users of debris-flow data.*
*Stage data are averaged over the recording period: the resulting approximations are negligible for the debris flows recorded since 1996 (recording intervals of 1 or 2 s), whereas they could be more relevant for the debris flows recorded in the first years (recording interval 10 s).*
*The start of a surge is easily identified at the first rise of the hydrograph; larger uncertainties can arise regarding the end of the recession phase, which occurs when the flow level becomes almost stable or a new surge begins. The process of surge identification is similar to hydrograph separation for water floods, although the sudden variations in the stage of debris flows make it somewhat more complicated. In the Moscardo the water level before the occurrence of a debris flow is negligible if compared to the maximum depth of debris-flow surges: as a consequence, there is no need to subtract a "baseflow" from the recordings of the debris-flow stage.*
*Although the surface of a debris flow is not perfectly planar, video recordings do not show remarkable differences in flow stage along a cross-section: one stage sensor at each instrumented cross-section is considered adequate to monitor debris-flow hydrographs.*
*Several factors influence the uncertainties of debris-flow measurements. Among the variables considered in this study, the debris-flow volumes are affected by the largest uncertainties because the assessment of volumes includes errors in flow depth measurement, approximations in the identification of the end of the surge(s), and possible variations in the geometry of the cross-section. A systematic analysis of uncertainties has not been carried out in the Moscardo. The experience from another instrumented catchment (Coviello et al., 2020), in which debris-flow volumes computed from the analysis of the hydrographs can be compared with debris*

*volumes accumulated in a sediment trap, shows that uncertainty in debris-flow volume can reach ± 50% for small events, i.e. debris flows with low flow depth.*

Line 83... (surge) velocity, mean velocity (see table 2): How is mean velocity calculated, which difference is calculated to (surge) velocity? How is the peak discharge really estimated?
*The methods for the computation of surge velocity, peak discharge, and volume are described in the submitted manuscript (section 3):*

> *"The mean debris-flow velocity was calculated as the ratio of the distance between two instrumented cross-sections to the time difference between the occurrence of the peak of the debris flow in the two recorded hydrographs. The debris-flow volume was computed by summing up, over the entire duration of the event, the product of mean flow velocity and cross-section area occupied by the flow at each time increment. The assumptions underlying this approach to volume computation, and the possible associated errors are discussed in Marchi et al. (2002) and Arattano et al. (2015)."*

Ch 4 (Line 100 ) ...beginning of summer (2019) to early autumn (1991). What does this mean? Are the triggering rainstorms independent from the gauging station?
*The earliest debris flow occurred at the beginning of summer (4 June 2019), the latest at the beginning of autumn (30 September 1991).*

Fig. 4 Legend is missing Are the data shown for all stations? What does 150 and 275present? Give the information about the day.
*Thank you for noticing the missing legend. When multiple rain gauges were working, only the one with the longest time series was selected for this figure. 150 and 275 are the day number of the year.*

Fig. 5: There is no significant regression! Why to present a regression? It is better to show the scattered data.
*We agree with this comment. We will modify the figure accordingly if we will be invited to submit a revised paper.*

L135 ...evacuation of sediment.. better: mobilization of sediment
*The terms "evacuation of sediment" or "sediment evacuation" are widely used to describe the export of sediment from a geomorphic system.*

Ch 4.2: For the reader it would be better to combine this Chapter with Ch 4.1 (Ocurrence). Well, Ch 4.1 shows the distributiion of df ocurrence during the year and Ch4.2 is focused on the precipitation thresholds, but there should be a link between the chapters to come out with some new findings. How is the duration of triggering rainfall defined? Is it the time before the debris flow arrives at the station or less?
*We will take these comments into account to modify the structure of section 4.*
*Possible revision:*
*4.1 Rainfall thresholds*
*4.2 Debris-flow occurrence*

*4.3 Debris-flow hydrographs*
*Duration and mean intensity of triggering rainstorms were computed from the onset of precipitation to the passage of the debris flows at the stage measurement stations. We plan to take into account also non-triggering rainstorm and to plot them in a duration-intensity plot. We wish to stress, however, that the automatic extraction of rainstorm events leads to the identification of duration and rainfall quantities that can hardly be compared to the expert-driven event identification. While the expert-based event definition can leverage the availability of debris-flow timing information and unravel the role and importance of rain and hiatuses, the automatic procedure relies only on thresholds of rainfall amount and intensity, showers separation. As such, on average the automatically-extracted rainfall events tend to be longer than the expert-identified ones as they include rainstorm tails.*

Table 1: Just a question: How do we define a catchment (area)? It seems that this area is calculated as the area of the drainage basin (which is hydrological defined). Usually a catchment area includes the are of the fan, too. (see Fig.1) How is mean basin slope and mean channel slope calculated?
*We respectfully disagree with the Reviewer about the computation of the catchment area. A drainage basin is the entire area providing runoff to a stream. A fan does not provide runoff to the stream, rather it is an area where flow (and sediment) divergence occurs.*
*Mean basin slope and mean channel slope were originally computed on a contour lines topographic map at the scale of 1:10,000: the resulting values are consistent with the computation on a catchment DEM.*

Table 2: mean velocity ???? (see above)
*See the answer to previous comments.*

Fig. 6: Please include the years of missing data in a different way, not only showing a Zero-value.
*There are no years with missing data on debris-flow occurrence. Even when the monitoring instrumentation was not working, the debris-flow occurrence was documented through field observations.*

Fig. 7: Legend is missing
*Thank you for this comment. We will add the legend if we will be invited to submit a revised paper.*

*Reference in this response*

*Coviello, V., Theule, J.I., Crema, S., Arattano, M., Comiti, F., Cavalli, M., Lucía, A., Macconi, P., Marchi, L., 2020. Combining Instrumental Monitoring and High-Resolution Topography for Estimating Sediment Yield in a Debris-Flow Catchment. Environmental & Engineering Geoscience, Vol. XXVI, No. 4, in press.*